# Multiple Dirac cones at the surface of the topological metal LaBi

Jayita Nayak[1,*], Shu-Chun Wu[1,*], Nitesh Kumar[1], Chandra Shekhar[1], Sanjay Singh[1], Jörg Fink[1,2], Emile E.D. Rienks[2,3], Gerhard H. Fecher[1], Stuart S.P. Parkin[4], Binghai Yan[1,5] & Claudia Felser[1]

The rare-earth monopnictide LaBi exhibits exotic magneto-transport properties, including an extremely large and anisotropic magnetoresistance. Experimental evidence for topological surface states is still missing although band inversions have been postulated to induce a topological phase in LaBi. In this work, we have revealed the existence of surface states of LaBi through the observation of three Dirac cones: two coexist at the corners and one appears at the centre of the Brillouin zone, by employing angle-resolved photoemission spectroscopy in conjunction with *ab initio* calculations. The odd number of surface Dirac cones is a direct consequence of the odd number of band inversions in the bulk band structure, thereby proving that LaBi is a topological, compensated semimetal, which is equivalent to a time-reversal invariant topological insulator. Our findings provide insight into the topological surface states of LaBi's semi-metallicity and related magneto-transport properties.

[1] Max Planck Institute for Chemical Physics of Solids, Nöthnitzer Str. 40, D-01187 Dresden, Germany. [2] Leibniz Institut für Festkörper- und Werkstoffforschung IFW Dresden, D-01171 Dresden, Germany. [3] Institute of Solid State Physics, Dresden University of Technology, Zellescher Weg 16, 01062 Dresden, Germany. [4] Max Planck Institute for Microstructure Physics, Weinberg 2, D-01620 Halle (Saale), Germany. [5] Max Planck Institute for Physics of Complex Systems, Nöthnitzer Str. 38, D-01187 Dresden, Germany. * These authors contributed equally to this work. Correspondence and requests for materials should be addressed to B.Y. (email: yan@cpfs.mpg.de) or to C.F. (email: felser@cpfs.mpg.de).

One of the most important fingerprints of a topological state of matter is a topological surface state (TSS). Topological materials include topological insulators (TIs)[1,2] and topological nodal semimetals, that are Dirac and Weyl semi-metals[3–8]. A TSS of a TI is commonly observed as a Dirac-cone type dispersion inside an insulating bulk energy gap[9–11], while a TSS of a Dirac or Weyl semimetal is characterized by Fermi arcs[12–16]. However, it is challenging to identify the topological nature of surface states for a family of gapless TIs that are characterized by the non-trivial $Z_2$ type topological invariants, dubbed $Z_2$-topological metals, due to the lack of a bulk energy gap. For instance, Dirac-like surface states have been found to overlap strongly with bulk states below the Fermi energy in the gapless Heusler TI compounds[17]. Only recently, the well known Rashba surface states of the element Au have been identified as TSSs[18]. The family of rare-earth monopnictides La$X$ ($X = $ P, As, Sb, Bi) can also be classified as $Z_2$-topological metals based on band structure calculations[19]. Moreover, a very large, unusual magnetoresistance has been observed in LaSb[20], LaBi[21] and a similar compound YSb[22], thus stimulating interest in directly observing any TSSs. By contrast, angle-resolved photoelectron spectroscopy (ARPES) on LaSb has revealed that this compound has a topologically trivial band dispersion[23].

LaBi is the compound with the strongest spin–orbit coupling in the family of rare-earth monopnictides. In this paper, we have investigated its TSSs by ARPES and *ab initio* calculations. Three Dirac cones have been identified in the surface band structure, unambiguously validating the topologically non-trivial nature of LaBi.

## Results

**Topology of the band structure.** LaBi crystallizes in the simple rock-salt structure (space group $Fm\bar{3}m$, No. 225), as shown in Fig. 1a. Figure 1b shows the bulk Brillouin zone (BZ) and the (001) projected two-dimensional surface Brillouin zone of the fcc lattice. $X$ and $L$ points of bulk BZ are projected to $\bar{M}$ and $\bar{X}$ points of the surface BZ. In the bulk band structure, the conduction and valence bands exhibit opposite parities of wave functions and get inverted near the $X$ point[19]. Although an indirect energy gap is missing with large electron and hole pockets at the Fermi energy[21], the direct energy gap still appears at every $k$-point, allowing us to define the topological $Z_2$ invariant. We found that La-$d$ and Bi-$p$ states contribute to the band inversion (see Supplementary Fig. 1). Three band inversions lead to a nontrivial $Z_2$ index $v_0 = 1$, which is consistent with calculations of the parity product of all valence bands at eight time reversal invariant $k$-points[24] that include the $\Gamma$ point, three nonequivalent $X$ points and four nonequivalent $L$ points. When these three band inversions are projected from the bulk to the (001) surface, three Dirac-cone like surface states appear. As shown in Fig. 1b, two non-equivalent $X$ points are projected to an $\bar{M}$ point in the surface BZ and the $\Gamma$ point is mapped to the surface $\bar{\Gamma}$ point. Therefore, as illustrated in Fig. 1c, we expect that two Dirac cones coexist at $\bar{M}$, where a direct bulk energy gap accidentally appears, and the third Dirac cone is located at $\bar{\Gamma}$, but fully overlaps with the bulk bands. The calculated band structures of the the LaBi (001) surface are shown in Fig. 1c,d, where the bright red lines represent the surface states. The Dirac points are marked by DP.

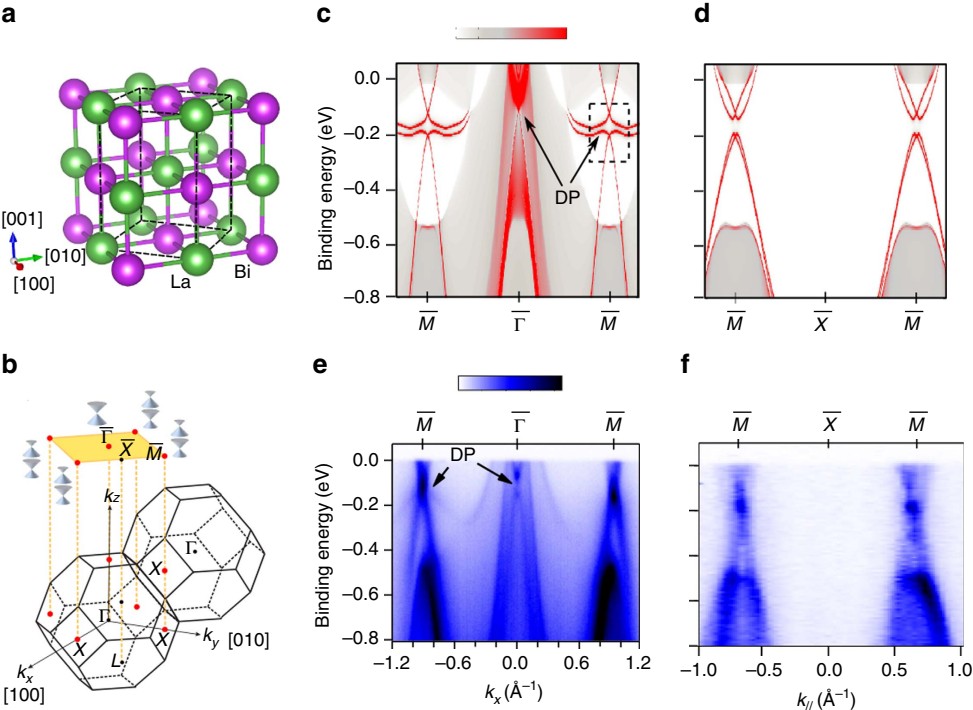

**Figure 1 | Fermi surfaces and the Dirac cones.** (**a**) Crystal structure of LaBi possessing simple NaCl type structure. (**b**) Bulk Brillouin zone of LaBi and the projection of the (001) surface Brillouin zone. In bulk, the band inversion occurs at the $X$ point. Two bulk $X$ points are projected to the surface $\bar{M}$ point while one bulk $\Gamma$ point is projected to the surface $\bar{\Gamma}$ point. Thus, two Dirac-cone-like surface states are expected to exist at $\bar{M}$ and one Dirac-cone-like surface state at $\bar{\Gamma}$, as is illustrated in graph (b) showing the surface Brillouin zone. (**c**) Calculated (001) surface band structure of LaBi along the $\bar{M} - \bar{\Gamma} - \bar{M}$ line providing the existence of a Dirac point (DP) at $\bar{\Gamma}$ and two Dirac nodes at $\bar{M}$. (**d**) Calculated surface band structure along the $\bar{M} - \bar{X} - \bar{M}$ line. (**c,d**) The colour bar from red to grey represents high to low surface contribution. (**e**) ARPES surface spectra measured along $\bar{M} - \bar{\Gamma} - \bar{M}$ direction with 83 eV photon energy. (**f**) ARPES surface spectra measured along $\bar{M} - \bar{X} - \bar{M}$ direction with 110 eV photon energy. (**e,f**) The colour bar from blue to white represents high to low intensity (a.u.).

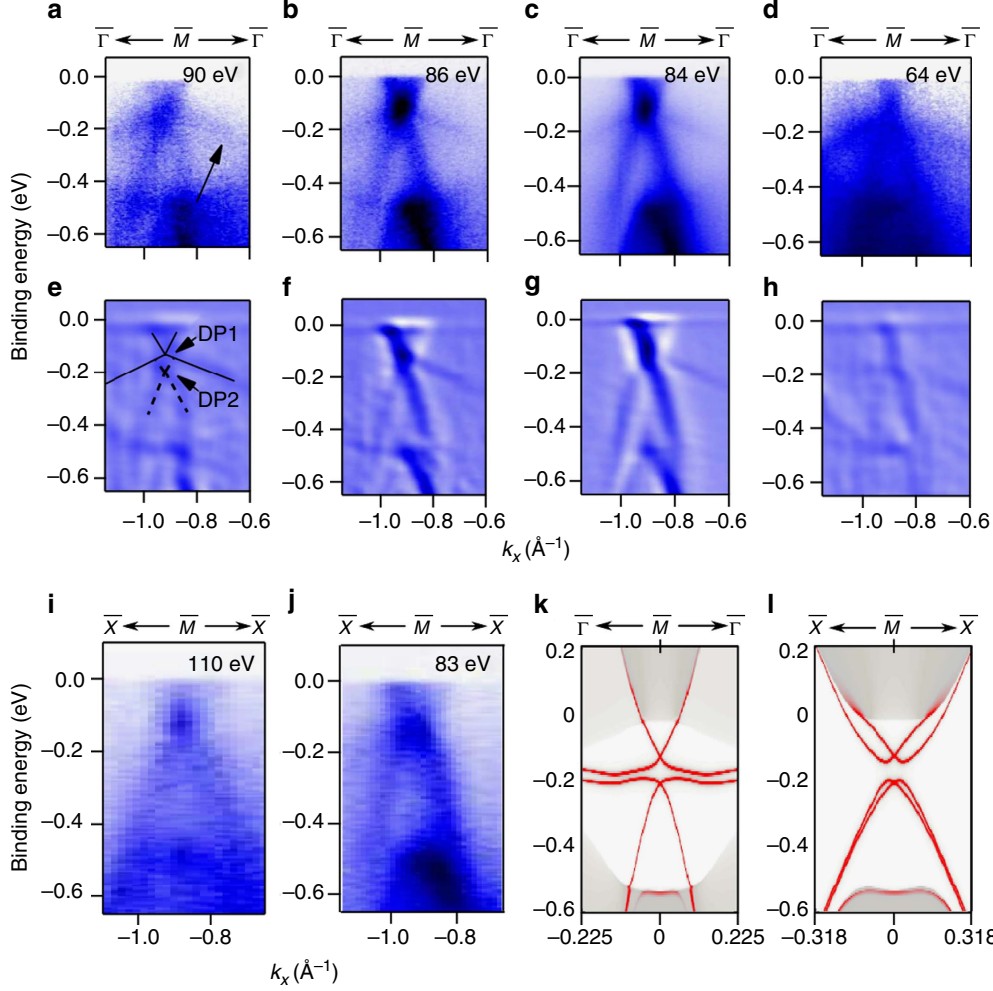

**Figure 2 | Strongly anisotropic Dirac cones at the $\bar{M}$ point. (a–h)** Zoomed images of the Dirac point at $\bar{M}$ at different photon energies (**a–d**) along $\bar{\Gamma} - \bar{M} - \bar{\Gamma}$ and its second derivative (**e–h**) to identify two Dirac points. (**i,j**) Dirac point at $\bar{M}$ along $\bar{X} - \bar{M} - \bar{X}$ with photon energies 110 and 83 eV, respectively. (**k,l**) Calculated band structure along $\bar{\Gamma} - \bar{M} - \bar{\Gamma}$ and $\bar{X} - \bar{M} - \bar{X}$, respectively. The bright red lines indicate the surface states while the grey regions represent bulk states.

**Surface states measured in experiment.** In ARPES, the Dirac point at $\bar{\Gamma}$ appears 150 meV below the Fermi energy whereas the surface states at $\bar{M}$ are split into two Dirac points with energies of 123 and 198 meV below $\epsilon_F$. Figure 1e,f show the ARPES spectra measured along the $\bar{M} - \bar{\Gamma} - \bar{M}$ and $\bar{M} - \bar{X} - \bar{M}$ directions, respectively, which are consistent with theoretical calculations. Details of the electronic structure around the $\bar{M}$ point have been investigated by varying photon energy and the results are shown in Fig. 2. The upper two rows show the spectra and their second derivatives along the $\bar{\Gamma} - \bar{M} - \bar{\Gamma}$ direction for photon energies from 64 to 90 eV. The lower panels (Fig. 2i) and (Fig. 2j) show spectra measured along the $\bar{X} - \bar{M} - \bar{X}$ direction for photon energies of 110 and 83 eV, respectively. For comparison, the calculated band structure is shown in panels (Fig. 2k) and (Fig. 2l) for the $\bar{\Gamma} - \bar{M} - \bar{\Gamma}$ and $\bar{X} - \bar{M} - \bar{X}$ directions, in which the bright red lines stand for surface states and the grey shadowed region stands for the bulk bands. The ARPES and calculations agree quite well. The surface states are anisotropic and differ for the two directions from $\bar{M}$. The horizontal parts of the TSSs appear only along $\bar{\Gamma} - \bar{M} - \bar{\Gamma}$. The separation of the Dirac points is only about 75 meV in experiment. The splitting is clearly resolved at 90 eV as is seen from the second derivative (Fig. 2e). These two Dirac-cone-like states remain at the same position in the band structure for different photon energies, indicating their surface state nature.

**Photon-energy-dependent ARPES.** To further validate the surface nature of the Dirac states, we have conducted photon-energy-dependent ARPES measurement (Fig. 3). The surface states do not disperse with photon energy (i.e., $k_z$), in contrast to the bulk states. Photoemission spectra recorded at various photon energies from 60 to 94 eV (Fig. 3a–i) reveal that both the Dirac cones at $\bar{\Gamma}$ and $\bar{M}$ do not disperse with photon energies. The resulting intensity distribution $I(k_x, k_z)$ and the dispersion $E(k_z)$ are shown in Fig. 3j,k, respectively. The white vertical lines in Fig. 3j mark the intensity of the top Dirac node observed at $\bar{M}$ and confirms the surface nature of the bands since there is no $k_z$ dependence throughout the whole energy range. The red horizontal line in Fig. 3k shows the position of the Dirac point at $\Gamma$. The binding energy of the Dirac point at $\bar{\Gamma}$ does not exhibit any $k_z$ dependence, unambiguously confirming the surface nature of the discussed Dirac bands. We have further studied another related compound GdSb which is antiferromagnetic below 20 K. The ARPES measurement at 1 K reveals that there is no Dirac surface state at the $\bar{\Gamma}$ and $\bar{M}$ points (see Supplementary Figs 2 and 3). It is further verified by our *ab initio* calculations where there is no band inversions occurring in the bulk band structure (see Supplementary Fig. 4). Therefore, we conclude that GdSb is topologically trivial.

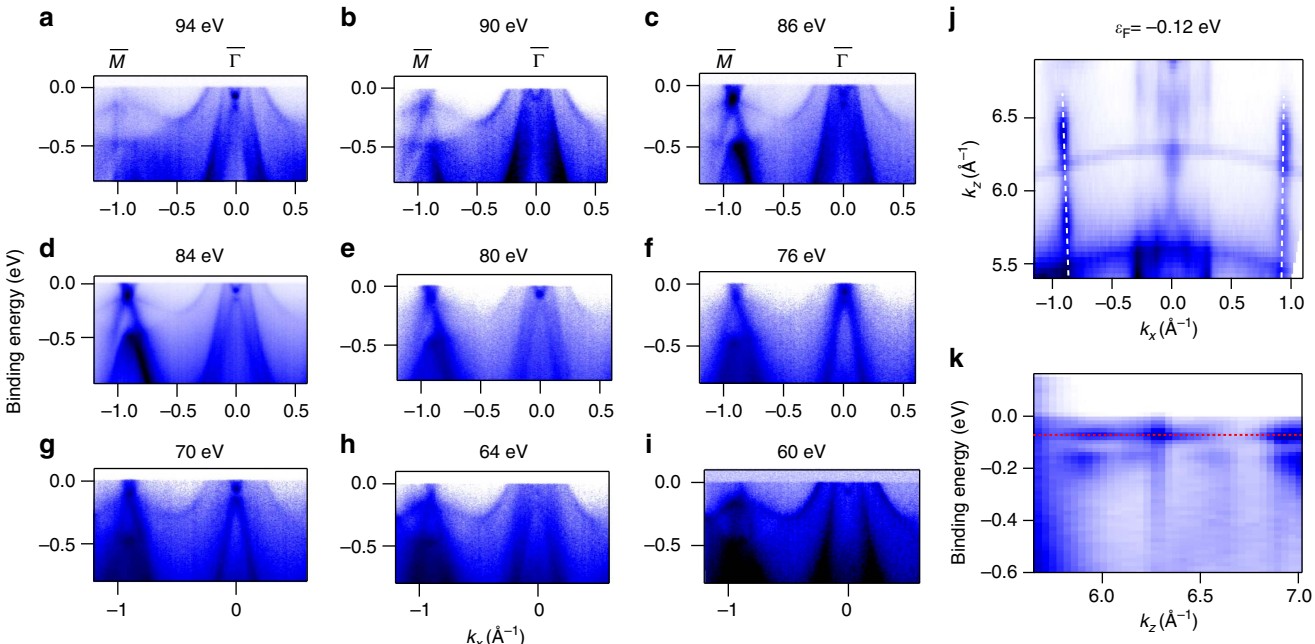

**Figure 3 | Confirmation of the topological surface state from the photon-energy-dependent ARPES.** (**a–i**) Photoemission intensity plots along $\bar{M}-\bar{\Gamma}$ direction at photon energies ranging from 94 to 60 eV with horizontal polarization. (**j**) ARPES spectral intensity maps in the $k_x-k_z$ plane at the binding energy of $-0.12$ eV, that is, at the top Dirac point at $\bar{M}$ indicating no $k_z$ dispersion of the Dirac node (shown by white dotted lines). (**k**) $E-k_z$ intensity maps providing no evidence of $k_z$ dispersion of the Dirac point at $\bar{\Gamma}$ (shown by red dotted lines).

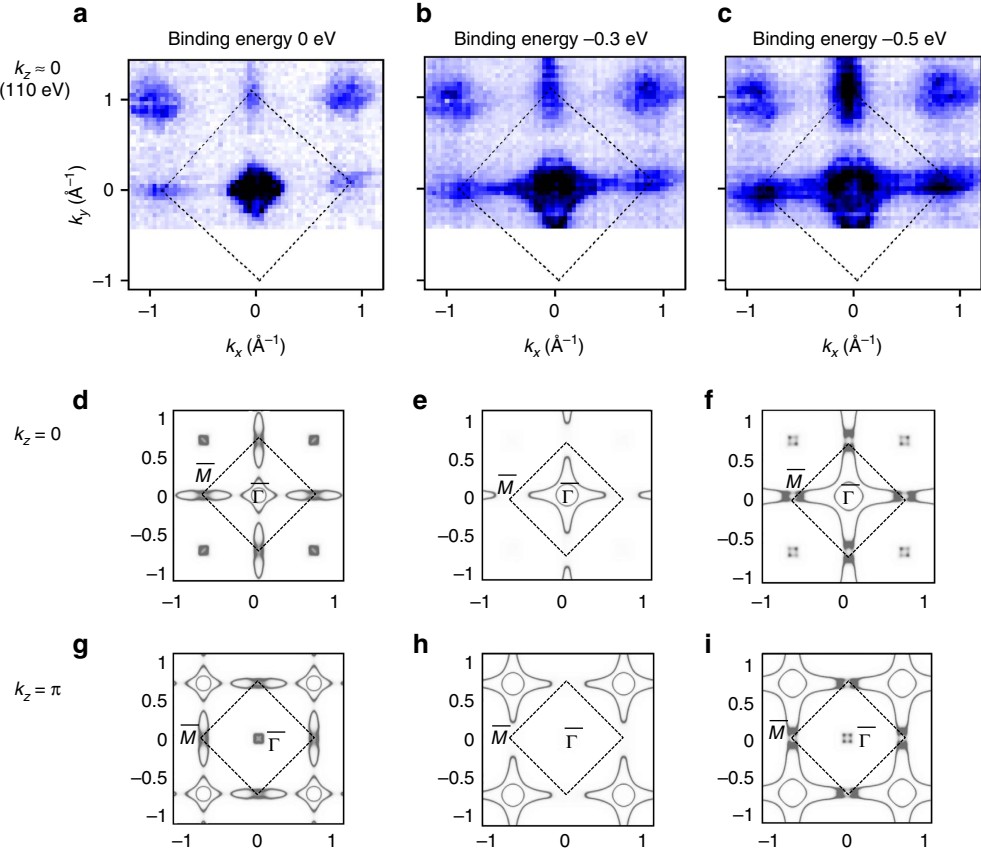

**Figure 4 | The Fermi surface of LaBi.** (**a–c**) ARPES Fermi surface measured with a photon energy of 110 eV, corresponding to the bulk $k_z \approx 0$ plane. (**d–i**) Calculated Fermi surfaces for $k_z = 0$ and $k_z = \pi$ planes, respectively. The different columns stand for different binding energies: 0 (**a,d,g**), $-0.3$ (**b,e,h**) and $-0.5$ (**c,f,i**) eV.

**Bulk Fermi surfaces**. Although the Dirac-cone-type surface states can be revealed in the ARPES band structures, signals of bulk states are still dominant in the ARPES results. It is interesting to observe the bulk bands from the Fermi surfaces. Different from surface states, the bulk bands exhibit strong $k_z$ dependence. Therefore, the Fermi surfaces look very different for different photon energies used in ARPES. According to our calculations (see Fig. 4), the Fermi surface looks like a cross centred at the $\bar{\Gamma}$ point of the first BZ for $k_z = 0$. However, at $k_z = \pi$ the Fermi surface exhibits a shift to the second BZ, leaving the $\bar{\Gamma}$ point of the first BZ relatively empty. Figure 4a–c shows the Fermi surface measured by ARPES for $k_z \approx 0$, which is well consistent with our calculations. Moreover, with decreasing the binding energy, one can find that the hole pockets at $\bar{\Gamma}$ increase in size while the electron pockets at $\bar{M}$ decrease in size. We note that the signals of surface state are too weak to be resolved in these Fermi surfaces. Additionally, it is worth mentioning that our experimental findings about LaBi can be seen in a broader context of TIs with NaCl structure, where AmN and PuTe were predicted to be correlated TIs[25].

After finishing the manuscript, we realized recent APRES measurements on NdSb[26], CeBi[27] and LaBi[28].

## Methods

**Single crystal growth and ARPES measurement.** LaBi single crystals were grown in a Bi flux[21] and the crystal structure was determined by X-ray diffraction using a four circle diffractometer. The ARPES measurements were carried out at the UE112-PGM2b beamline of the synchrotron radiation facility BESSY (Berlin) using the $1^3$-ARPES end station that is equiped with a Scienta R4000 energy analyzer. All measurements were performed at a temperature of 1 K at various photon energies from 50 to 110 eV using both horizontal and vertical polarizations. The total energy resolution was approximately 4 meV and the angular resolution was 0.2°.

**Band structure calculations.** The electronic structure calculations were carried out using the local density approximation of the density-functional theory as implemented in the Vienna ab initio simulation package (VASP)[29]. The generalized gradient approximation[30] was employed for the exchange-correlation energy functional. Spin–orbit interaction was included as pertubation. The surfaces states were calculated by projecting the density of states to the top four atomic layers of a half-infinite surface with the Green's function method based on the tight-binding parameters from Wannier functions[31].

**Data availability.** The data that support the findings of this study are available from the corresponding authors B.Y. and C.F. upon request.

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

## Acknowledgements

This work was financially supported by the the ERC (Advanced Grant No. 291472 Idea Heusler).

## Author contributions

J.N., J.F., E.E.D.R. and G.H.F. performed ARPES experiments. S.-C.W. and B.Y. performed *ab initio* calculations. N.K. and C.S. grew single crystals. B.Y. wrote the manuscript with helpful inputs from J.N., S.-C.W. and J.F. All Authors contributed to scientific discussions.

## Additional information

**Competing financial interests:** The authors declare no competing financial interests.

**Publisher's note**: 

