## [Peer Review File · Nature Communications]

Reviewers' comments:

Reviewer #1 (Remarks to the Author):

The search for novel topological states of matter such as topological semimetal/metal becomes one of the central themes in condensed matter physics and materials sciences. Identifying the topological nature of surface states for these gapless systems is a challenging and important issue in experiments. By combining ARPES and ab initio calculations, the authors revealed the existence of surface states in the semimetal LaBi, with three Dirac cones in its surface Brillouin zone (BZ), and they recognize the topological origin of these Dirac surface states as the Z_2 -type topological band inversion. The paper is well written and the results are reliable, timely and interesting. I thus recommend the publication of this paper. Meanwhile, there are also a few minor points for the authors to consider:

- 1) Seen from Fig. 1(c), the Dirac point at $\bar{\Gamma}$ is below both two Dirac points at \bar{M} . However, the authors claimed in the text that "The Dirac point at $\bar{\Gamma}$ appears 150 meV below the Fermi energy whereas the surface states at \bar{M} are split into two Dirac points with energies of 123 and 198 meV below ϵ_F ", i.e., the Dirac point at $\bar{\Gamma}$ is between the two Dirac points at \bar{M} in energy. Is this the experimental result [Fig. 1(e)]? The authors should make it clear.
- 2) Please check the label " k_x " in Figs. 4(b) and 4(c). Should one " k_x " be changed by " k_y "?
- 3) Reference 28 is not consistent with the text content part. Please double check it.
- 4) The two Dirac points at \bar{M} in the surface BZ originate from the bulk band inversions at two X points. In the present paper for LaBi, they are split in energy; however, in reference 27 for CeSb, they are degenerate. How to understand this?

Reviewer #2 (Remarks to the Author):

This is an important and interesting new compound, and it is nice to see the collaboration between the theory and experimental group on this work. However, I have a few important concerns so the paper is not obviously sound nor publishable at present.

The main point of the paper is that there is a single surface Dirac cone at Γ and two Dirac cones at the \bar{M} points in LaBi, summarized in fig 4d. They also studied a related compound GdSb "we found a gap in the surface states at M point, indicating that GdSb is topologically trivial." Their metric appears to be: "If they find a gap at the DP it is topologically trivial. If they find no gap then it is a TI." On the other hand, the two claimed DPs at \bar{M} in LaBi have a gap between them - how do we distinguish this gap in LaBi from what they see in GdSb? From the experimental data the two compounds are not so obviously different. Are they different in theory? No theory was shown for GdSb.

Their description of the Fermi surface in Fig 4 is very unclear, both theoretically and experimentally. They discuss the cigar shaped pockets at the \bar{M} points, but it wasn't fully clear whether these are surface or bulk, or how they relate to the Dirac cones. Panel f of the same panel is all about Dirac points so I assume they connect and they argue that these Fermi surfaces are from the Dirac points, but that was not made clear in the text.

They claim that the Dirac points at M should have an unusual behavior. The calculated ones in Fig 2c have a very different behavior than the schematic in fig 4d. How should we understand this difference? In what way are the things shown in Fig 2c "Dirac"? Are the details (such as for example the gap size between them) sensitive to the thickness of the slab or the vacuum layer used to make the calculations? How much shall we trust the separation of bulk and surface in the calculations where the bulk states are shaded? (Often I see the results of such calculations as showing a close superposition of many states due to a very tall unit cell with the "bulk" the fine superposition of these states. Did the authors just shade in the regions between these states? With what precision can that be done?)

REVIEWERS' COMMENTS:

Reviewer #1 (Remarks to the Author):

The authors have clarified all the scientific concerns addressed in my previous report. Therefore, I recommend its publication as is.

Reviewer #3 (Remarks to the Author):

This work reports a combined first principles calculation and ARPES study of LaBi, argued to be a Z2 topological metal. A Z2 topological metal is basically the same as a Z2 topological insulator - there is a direct bulk gap throughout the Brillouin zone and protected surface states cross this gap. The difference is that the bulk bands cross the Fermi level at some momentum points in the former case making it a bulk metal. The key to recognizing LaBi as a Z2 topological metal is through the counting of surface states as stated by the authors. The calculated band structure shown in figure 1c is topological because neither of the two Dirac cones at M-bar connect the bulk conduction and valence bands while the Dirac cone at Gamma-bar does connect them, leading to only one (odd number) protected cone. The ARPES data and calculation agree well at Gamma-bar, the crucial question is whether they agree at M-bar. By varying the photon energy, the authors make a convincing case distinguishing the surface and bulk states at M-bar. However, the resolution of the data is far too coarse to confirm the intricate surface-to-bulk connectivity shown in the calculation at M-bar. Furthermore, to firmly establish an odd versus even number of cones, both inequivalent M-bar points need to be studied in detail, not just one of them as done in figure 3 and 4. In fact there is not complete equivalence between the two independent M-bar points as seen in figure 1e and this is not addressed in the manuscript. Therefore I think the data can be interpreted as being consistent with the calculation on a qualitative level, but the data is not of sufficient quality to independently confirm LaBi as a Z2 topological metal. Claiming this is experimental evidence proving LaBi to be a Z2 topological metal is too strong in my opinion. The work overall is technically sound and I think the authors have adequately addressed the previous reviewer questions. But for these reasons mentioned above, I think the work is more suitable for an archival journal.

Reply to Reviewers' comments:

-----Reviewer #1 (Remarks to the Author):-----

Comment: *The search for novel topological states of matter such as topological semimetal/metal becomes one of the central themes in condensed matter physics and materials sciences. Identifying the topological nature of surface states for these gapless systems is a challenging and important issue in experiments. By combining ARPES and ab initio calculations, the authors revealed the existence of surface states in the semimetal LaBi, with three Dirac cones in its surface Brillouin zone (BZ), and they recognize the topological origin of these Dirac surface states as the Z₂-type topological band inversion. The paper is well written and the results are reliable, timely and interesting. I thus recommend the publication of this paper. Meanwhile, there are also a few minor points for the authors to consider:*

Reply: We appreciate the positive comments from the Reviewer. In the following we have addressed all reviewer's questions.

Comment: *1) Seen from Fig. 1(c), the Dirac point at $\bar{\Gamma}$ is below both two Dirac points at \bar{M} . However, the authors claimed in the text that "The Dirac point at $\bar{\Gamma}$ appears 150 meV below the Fermi energy whereas the surface states at \bar{M} are split into two Dirac points with energies of 123 and 198 meV below e_F ", i.e., the Dirac point at $\bar{\Gamma}$ is between the two Dirac points at \bar{M} in energy. Is this the experimental result [Fig. 1(e)]? The authors should make it clear.*

Reply: We thank the referee for this professional question. In the tight-binding surface state calculations, the surface local potential is usually different from the real surface. Thus, the surface state dispersion can be slightly different from ARPES, although the topology of surface states remains the same. In the modified manuscript, we have optimized the tight-binding parameters and plotted all surface states again. Now the calculation results agree quantitatively well with ARPES, which still preserves the same topology.

Comment: *2) Please check the label " k_x " in Figs. 4(b) and 4(c). Should one " k_x " be changed by " k_y "? 3) Reference 28 is not consistent with the text content part. Please double check it.*

Reply: We have corrected the typos and the reference 28.

Comment: *4) The two Dirac points at \bar{M} in the surface BZ originate from the bulk band inversions at two X points. In the present paper for LaBi, they are split in energy; however, in reference 27 for CeSb, they are degenerate. How to understand this?*

Reply: It is true that the Dirac points at M split for LaBi, but not (or quite tiny) for CeSb. However, whether they split or not will not change the topology – the odd number of Dirac cones. The splitting of LaBi was also verified by a high-resolution ARPES experiment recently [arXiv:1607.04178]. In addition, the local surface potential can sensitively tune the coupling between two Dirac points. As discussed in Fig. S10 of ref.27 [arXiv:1604.08571], for example, the splitting of two Dirac points can be tuned by the positions of the top atomic layer.

-----Reviewer #2 (Remarks to the Author):-----

Comment: *This is an important and interesting new compound, and it is nice to see the collaboration between the theory and experimental group on this work. However, I have a few important concerns so the paper is not obviously sound nor publishable at present.*

Reply: We thank the Reviewer for his/her review on our manuscript.

Comment: *The main point of the paper is that there is a single surface Dirac cone at gamma and two Dirac cones at the Mbar points in LaBi, summarized in fig 4d. They also studied a related compound GdSb "we found a gap in the surface states at M point, indicating that GdSb is topologically trivial." Their metric appears to be: "If they find a gap at the DP it is topologically trivial. If they find no gap then it is a TI." On the other hand, the two claimed DPs at Mbar in LaBi have a gap between them - how do we distinguish this gap in LaBi from what they see in GdSb? From the experimental data the two compounds are not so obviously different. Are they different in theory? No theory was shown for GdSb.*

Reply: We explain the difference between LaBi and GdSb more comprehensively, rewrite the text in SI that is misleading and add the GdSb band structures in the SI.

The criteria to identify the topology are to count the odd number of Dirac-cone-like surface states measured by ARPES, and the odd number of band inversions in the bulk band structure. In ARPES, there are three Dirac cones for LaBi: two at the \bar{M} point and one at the $\bar{\Gamma}$ point. In contrast, there are no Dirac states at \bar{M} and $\bar{\Gamma}$ points for GdSb. In the calculated bulk band structure, LaBi exhibits three times of band inversions between the conduction and valence bands at three nonequivalent X point. However, GdSb exhibits no band inversion with a trivial Z2 index calculated from the parity (Supplementary Figure 4). Therefore, we conclude that LaBi is a topological metal while GdSb is a trivial metal.

Comment: *Their description of the Fermi surface in Fig 4 is very unclear, both theoretically and experimentally. They discuss the cigar shaped pockets at the Mbar points, but it wasn't fully clear whether these are surface or bulk, or how they relate to the Dirac cones. Panel f of the same panel is all about Dirac points so I assume they connect and they argue that these Fermi surfaces are from the Dirac points, but that was not made clear in the text.*

Reply: We thank the Reviewer for pointing out this question. The surface states are overwhelmed by the bulk states in the Fermi surface mapping by ARPES. So the main feature of the original Figure 4 refers to the bulk states. To avoid the confusion, we move the illustrations of Dirac cones to the new Figure 1. We have made a new Figure 4 to demonstrate the kz and energy dependences of the bulk bands.

We have added the related discussion on Pages 4-5. "Although the Dirac-cone-type surface states can be revealed in the ARPES band structures, signals of bulk states are still dominant in the ARPES results. It is interesting

to observe the bulk bands from the Fermi surfaces. Different from surface states, the bulk bands exhibit strong k_z dependence. Therefore, the Fermi surfaces look very different for different photon energies used in ARPES. According to our calculations (see Figure 4), the Fermi surface looks like a cross centered at the $\overline{\Gamma}$ point of the first BZ for $k_z = 0$. However, the Fermi surface exhibits a shift to the second BZ, leaving the $\overline{\Gamma}$ point of the first BZ relatively empty. Figure 4a shows the Fermi surface measured by ARPES for $k_z \approx 0$, which is well consistent with our calculations. Moreover, with decreasing the Fermi energy, one can find that the hole pockets at $\overline{\Gamma}$ increase in size while the electron pockets at \overline{M} decrease in size. We note that the signals of surface state are too weak to be resolved in these Fermi surfaces.”

Comment: They claim that the Dirac points at M should have an unusual behavior. The calculated ones in Fig 2c have a very different behavior than the schematic in fig 4d. How should we understand this difference? In what way are the things shown in Fig 2c "Dirac"? Are the details (such as for example the gap size between them) sensitive to the thickness of the slab or the vacuum layer used to make the calculations? How much shall we trust the separation of bulk and surface in the calculations where the bulk states are shaded? (Often I see the results of such calculations as showing a close superposition of many states due to a very tall unit cell with the "bulk" the fine superposition of these states. Did the authors just shade in the regions between these states? With what precision can that be done?)

Reply:

- (1) In Figs. 1c-1d, we can see clearly surface states at the \overline{M} point. These surface bands cross each other at \overline{M} , presenting two crossing points. Here the band crossing points are protected by the time-reversal symmetry (TRS). Very close to each crossing point, the band dispersion is linear. In the sense of linearly crossing protected by TRS, these two band-crossing points at \overline{M} are well-defined Dirac points. We note that the real Dirac cones are very anisotropic, as shown in Figs. 1c-1d. Only for schematic purpose, we indicated two Dirac cones at \overline{M} in previous Fig.4d (the new Fig. 1b). We further clarified these questions in the modified manuscript.
- (2) Our calculated surface states are not related to the thickness of a slab model, because we did not employ a slab model, but a semi-infinite surface. However, the Reviewer’s question is still relevant in the semi-infinite surface model. In our calculations, we project all DOS of a semi-infinite surface to the surface atomic layers, as the local density of states (LDOS). LDOS includes both surface and bulk states together, without masking an extra bulk projection. Therefore, the thickness the surface layers will affect the intensity contrast between the surface and bulk states. If the surface layer is thicker, the bulk states look stronger. If the surface layer is thinner, the surface states look stronger, but their positions do not change. However, some bulk states become weaker or even disappear. Furthermore, it is usually not easy for ARPES to tell

exactly how deep it can see into the surface. Therefore, varying the thickness of the surface projection layer is a practical way to distinguish the surface and bulk states in calculations and interpret ARPES results. We show the LDOS for surface layers of 4 (one unit cell) and 8 (two unit cells) atomic-layer-thick in the following. It is clear that the surface bands are much sharper in the 4 atomic-layer-thick case [Fig.(a)], which is used as new Figs. 1c in the modified manuscript.

-----Reviewer #3 (Remarks to the Author) -----

This work reports a combined first principles calculation and ARPES study of LaBi, argued to be a Z2 topological metal. A Z2 topological metal is basically the same as a Z2 topological insulator - there is a direct bulk gap throughout the Brillouin zone and protected surface states cross this gap. The difference is that the bulk bands cross the Fermi level at some momentum points in the former case making it a bulk metal. The key to recognizing LaBi as a Z2 topological metal is through the counting of surface states as stated by the authors. The calculated band structure shown in figure 1c is topological because neither of the two Dirac cones at M-bar connect the bulk conduction and valence bands while the Dirac cone at Gamma-bar does connect them, leading to only one (odd number) protected cone. The ARPES data and calculation agree well at Gamma-bar, the crucial question is whether they agree at M-bar. By varying the photon energy, the authors make a convincing case distinguishing the surface and bulk states at M-bar. However, the resolution of the data is far too coarse to confirm the intricate surface-to-bulk connectivity shown in the calculation at M-bar. Furthermore, to firmly establish an odd versus even number of cones, both inequivalent M-bar points need to be studied in detail, not just one of them as done in figure 3 and 4. In fact there is not complete equivalence between the two independent M-bar points as seen in figure 1e and this is not addressed in the manuscript. Therefore I think the data can be interpreted as being consistent with the calculation on a qualitative level, but the data is not of sufficient quality to independently confirm LaBi as a Z2 topological metal. Claiming this is experimental evidence proving LaBi to be a Z2 topological metal is too strong in my opinion. The work overall is technically sound and I think the authors have adequately addressed the previous reviewer questions. But for these reasons mentioned above, I think the work is more suitable for an archival journal.

Our reply:

We thank the Reviewer for his/her review and comments on our manuscript.

The Reviewer questioned the existence of an odd number of Dirac cones. In particular he/she asks whether the ARPES data agree with the calculations at the M-bar point. The Reviewer comes to the conclusion that the energy “resolution is far too coarse” to resolve this issue. Furthermore the Referee points out that the dispersion at the two independent M-bar points may be different which questions our statement of an odd number of Dirac cones. In the following we answer these questions.

We agree with the Reviewer 3 that there are two M-bar points in the unit cell, \bar{M}_x and \bar{M}_y . We have analyzed in our experiment only one M-bar point, namely \bar{M}_y . On the other hand, we point out that a difference between \bar{M}_x and \bar{M}_y is not expected for a simple cubic rock-salt structure.

- (1) The experimental second derivative data in Fig. 2 (a) perfectly agree with the calculations and clearly show evidence for two Dirac points at the \bar{M}_y point. The agreement is also quite good for the other photon energies (86, 84, and 64 eV).
- (2) Finally from the argumentation of the Reviewer, one may conclude that our energy resolution of 4 meV is not state of the art in ARPES. We are aware that laser ARPES provides an energy resolution of 0.07 meV [Okazaki et al.

Science 337, 1314 (2012)]. The problem with this technique is that there is only one photon energy (7 eV). To discriminate between surface states and bulk states we need variable photon energies to obtain information on the k_z dependence. This can be only performed at synchrotron facilities which provide photons with variable photon energies. The energy resolution of such ARPES spectrometers is reduced compared to laser ARPES. In the following we present a compilation of the energy resolution at a photon energy of about 50 eV of various end stations which produced a large fractions of all present ARPES results.

Stanford Synchrotron Radiation source	10 meV
Advanced Light source Berkeley	20 meV
Diamond	5 meV
Swiss Light Source	4 meV
Elettra	6 meV

This compilation clearly shows that the energy resolution of our end station is state of the art and to our knowledge is at the lower edge of all synchrotron end stations. Furthermore we point out that according to our knowledge, the achieved lowest sample temperature of 1 K in our end station is not reached in any other ARPES spectrometer. This means that in our data there is essentially no additional thermal broadening which appears at the other end stations reaching sample temperatures above 10 K.